# Chitosan-Modified Biochars to Advance Research on Heavy Metal Ion Removal: Roles, Mechanism and Perspectives

**DOI:** 10.3390/ma15176108

**Published:** 2022-09-02

**Authors:** Justyna Bąk, Peter Thomas, Dorota Kołodyńska

**Affiliations:** 1Department of Inorganic Chemistry, Institute of Chemical Sciences, Faculty of Chemistry, Maria Curie-Skłodowska University, Maria Curie-Skłodowska Sq. 2, 20-031 Lublin, Poland; 2Earthcare, LLC, 8524 Southport Drive, Evansville, IN 47711, USA

**Keywords:** biochar composites, chitosan, heavy metal ions, sorption mechanism

## Abstract

The chitosan-modified biochars BC-CS 1-1, BC-CS 2-1 and BC-CS 4-1 were subjected to the synthetic application of biochar from agriculture waste and chitosan for the adsorption of Cu(II), Cd(II), Zn(II), Co(II) and Pb(II) ions from aqueous media. The results displayed a heterogeneous, well-developed surface. Additionally, the surface functional groups carboxyl, hydroxyl and phenol, determining the sorption mechanism and confirming the thermal stability of the materials, were present. The sorption evaluation was carried out as a function of the sorbent dose, pH, phase contact time, initial concentration of the solution and temperature. The maximum value of q_t_ for Pb(II)-BC-CS 4-1, 32.23 mg/g (C_0_ 200 mg/L, mass 0.1 g, pH 5, 360 min), was identified. Nitric acid was applied for the sorbent regeneration with a yield of 99.13% for Pb(II)-BC-CS 2-1. The produced sorbents can be used for the decontamination of water by means of the cost-effective and high-performance method.

## 1. Introduction

As the results of civilizational development and increasing industrialization, changes in the modern world greatly affect the natural environment, resulting in ecosystem degradation. Moreover, the insufficient treatment of industrial wastewater, inefficient technologies, waste gases and dust released into the environment increase the heavy metal concentrations in the natural environment [1]. These substances pose a danger to the health and life of humans and animals due to their tendency towards bioaccumulation and the fact that some of them are carcinogenic [2]. This is due to heavy metals’ disturbance of metabolic and cellular activities, forming links with nucleic acids and proteins of the thiol functional group [3]. Thus, it is important to remove the heavy metals excess from sewages in order to confine their displacement to other parts of the trophic chain [4]. As a result, numerous researchers have focused their attention on the methods of their removal from waters and wastewaters [5,6]. Adsorption is the most promising approach [7] because of its low cost, wide range of applications, adaptability and ease of use [2,8].

Therefore, raw materials, which are widely available in the environment and in industrial and agricultural waste, can be used as inexpensive sorbents. Using processing methods, these raw materials (applying a controlled process of thermal decomposition in the absence or with the confined access of oxygen) allow us to obtain a cheap biochar [9,10,11]. This makes plant- and animal-origin waste management possible, providing both ecological and economic benefits [12]. Variable parameters can be applied to characterize biochar depending on the substrates used in its manufacture, process conditions and prospective uses [13,14,15]. The stability, porosity and presence of the functional groups of the surfaces are obvious advantages of biochars [16]. As a result, Cu(II), Cd(II), Zn(II), Co(II) and Pb(II) are just a few of the heavy metal ions that biochars can capture from waters and wastewaters [17,18]. As compared to the activated carbons, biochars are cost-effective sorbents. The simple procedure for their acquisition has encouraged many researchers to make use of their potential for modification in order to enhance their qualities [19].

Chitosan, as one of the most common polymers found in the natural environment, is created as a consequence of chitin deacetylation [20]. The presence of intra- and inter-molecular hydrogen bonds in the lattice determines the ordered structure. Chitosan is characterized by its biodegradability, non-toxicity, low price and wide availability [21]. Furthermore, its optimal sorption capacity due to the presence of amino and hydroxyl groups facilitates its use in the removal of inorganic chemicals, antibiotics and dyes from waters and sewages [20,22,23]. The chitosan-modified biochar combines the positive qualities of both materials, as well as their low manufacturing costs [24].

The main objective of this research was to prove that chitosan-modified biochars are suitable sorbents for Cu(II), Cd(II), Zn(II), Co(II) and Pb(II) ion removal from aqueous media. In order to prove the above mentioned hypothesis, three biochar-based materials differing in the amounts of chitosan added during their processing were synthesized. The sorbents were designated as BC-CS 1-1, BC-CS 2-1 and BC-CS 4-1. Heavy metal ions kinetic, adsorption and desorption analyses were optimized and discussed. The process parameters were determined using the kinetic models (pseudo first order, pseudo second order and intraparticle diffusion), isotherm models (Langmuir, Freundlich, Temkin and Dubinin–Raduszkiewicz) and thermodynamic calculations to estimate the possible mechanism of the adsorbent–adsorbate interactions. For the determination of their properties, the sorbents were subjected to physicochemical characterization by means of CHN analysis, scanning electron microscopy (SEM), gas porosimetry (ASAP), X-ray diffractometry (XRD), Fourier-transform infrared spectroscopy using the attenuated total reflectance technique (ATR-FTIR), thermogravimetry (TG) and differential thermogravimetry (DTG). The potentiometric approach was also used to determine the point of zero charge and the density of the surface charge. The study of the structural and physicochemical properties of the sorbents BC-CS 1-1, 2-1 and 4-1, in conjunction with a full description of the kinetic, isotherm and thermodynamic parameters, is undoubtedly a novelty of this paper. Importantly, the research on the sorption and desorption cycles allows us to understand the complex mechanism of heavy metal ions removal from aqueous media.

## 2. Materials and Methods

### 2.1. Sorbent Preparation and Characterization

The biochar applied in this research was obtained from Earthcare (Evansville, IN, USA), taken from the agriculture waste in the gasification process. The chitosan used in the experiments was provided by Sigma-Aldrich (Poznań, Poland) and its powder form was applied without prior processing. A total of 2 g of chitosan was dissolved in 100 mL of 2% acetic acid solution and stirred for 24 h with a magnetic stirrer at 1000 rpm at room temperature. Then, the biochar was added to the formed mixture in the amount of 2 g for BC-CS 1-1, 4 g for BC-CS 2-1, 8 g for BC-CS 4-1, followed by stirring for 30 min. A total of 100 mL of 3% sodium hydroxide solution was added in the next stage to precipitate the suspension. Then, it was stirred for 2 h and washed with distilled water. When it was neutral, the mixture was filtered at a reduced pressure. The obtained material was dried in the laboratory oven at about 368 K for 8 h and then ground. The synthesis scheme is presented in Figure 1.

The CHN analyzer 2400 (PerkinElmer, Waltham, MA, USA) was used for the elemental analysis to calculate the percentage amounts of carbon, hydrogen and nitrogen. The ash content was determined using the laboratory oven Thermolyne (LaboPlus, Thermo Scientific, Waltham, MA, USA) at the temperature of 1088 ± 283 K.

In order to characterize the surface morphology of the biochar-based materials, scanning electron microscopy (SEM) analysis was performed using the Quanta 3D FEG (FEI, Hillsboro, OR, USA) equipment.

The low temperature N_2_ adsorption/desorption isotherms were determined using an ASAP 2420M (Micrometrics, Norcross, GA, USA) sorption analyzer. The BET surface area, pore diameter, total pore volume and micropore volume were recorded based on the ASAP analysis report. The pore size distribution was determined using the adsorption isotherms. To determine the pore size distribution, the BJH method was used. The BET ‘c’ constant varied with the surface area and was positive. A negative ‘c’ value arises from a negative intercept on the BET plot and is caused by the used data point being at P/P_0_ values that are too high.

The X-ray diffraction analysis (XRD) was performed using a PANalytical X-ray diffractometer (Empyrean, Dordrecht, the Netherlands) to identify the crystalline phases present in the sorbents.

The analysis of the infrared spectra was performed using the ATR technique, using a Cary 630 FTIR analyzer (Agilent Technologies, Santa Clara, CA, USA). The attenuated internal total reflection (ATR) technique is based on the total internal reflection of light. In the FTIR method, the measurement is made in the wavelength range of 4000–400 cm^−1^ and with a spectral resolution of 4 cm^−1^. By measuring the infrared spectra, it is possible to determine the functional groups present in the analyzed materials.

The thermal analysis was performed on a Q50 TGA apparatus (TA Instruments, New Castle, DE, USA). The test sample, weighing about 25 mg, was heated at a constant rate of 10 K/min in the temperature range of 298–1273 K in a nitrogen atmosphere. Simultaneously with the TG analysis, the DTG analysis was performed. As a result, in addition to the thermogravimetric curve, the first derivative of the thermogravimetric curve with respect to the temperature was obtained.

The point of zero charge (pH_PZC_) was determined by potentiometric titration. The measurement consisted of placing 0.5 g of sorbents in 50 mL of NaCl solution with various concentrations, including 0.1, 0.01 and 0.001 mol/L, and measuring the pH with a 907 Titrando potentiometric titrator (Metrohm, Herisau, Switzerland). Then, the sample was titrated with 1 mol/L of HCl and 1 mol/L of NaOH alternating solutions by dosing it with a constant volume of the titrant, and the pH value of the solution was automatically recorded after each dose of the titrant. Additionally, the pH_PZC_ values of the sorbents were determined by the drift method. For this purpose, 0.5 g of sorbents was shaken with 50 mL of NaCl solution in the pH in the range of 2–12 for 24 h. After this time, the changes in pH were measured and, based on the dependence of the ∆pH (pH_final_–pH_initial_, where pH_final_ is the pH value measured after shaking time, and pH_initial_ is the pH value of the initial solution) on the pH_initial_, the point of zero charge was read. The pH_PZC_ value is the pH value at which the sorbent is electrically neutral. Moreover, the density of the surface charge was calculated as:(1)$$\sigma =\frac{FC\Delta V}{mA}$$
where F is the Faraday constant [9.648·10^4^ C/mol], C is the concentration of acid/base solution used during the titration [mol/L], ∆V is the ratio of the volume of acid or base added during titration [L], m is the sample weight [g] and A is the specific surface area [m^2^/g].

### 2.2. Kinetic and Adsorption Investigations

Cu(NO_3_)_2_·3H_2_O, Cd(NO_3_)_2_·4H_2_O, Zn(NO_3_)_2_·6H_2_O, Co(NO_3_)_2_·6H_2_O and Pb(NO_3_)_2_ were applied as sources of heavy metal ions. These chemicals were dissolved in distilled water to create the stock solution. To achieve the desired pH, 1 mol/L HNO_3_ and/or 1 mol/L NaOH were added dropwise to the solution. All substances were of the greatest analytical quality. First, the dosage impact was investigated. The mixture included 0.1 g (5 g/L), 0.15 g (7.5 g/L) or 0.2 g (10 g/L) of the biochar-based sorbents and 20 mL of the Cu(II) ions solution. The samples were mixed at pH 5 for 360 min at room temperature in a mechanical shaker type 358 A (Elpin+, Katowice, Poland) at a constant vibration amplitude (8 units). The sorbent was removed from the solution by filtration after shaking, and the filtrate pH was determined using the pHmeter pHM82 (Radiometer, Copenhagen, Denmark). The filtrate M(II) ion concentration was studied using the atomic absorption analyzer Spectr AA240 FS (Varian, Palo Alto, CA, USA) at 324.7 nm for Cu(II), 228.8 nm for Cd(II), 213.9 nm for Zn(II), 240.7 nm for Co(II) and 217.0 nm for Pb(II). The appropriate sorbent mass was determined based on these findings.

To study the pH effect, 0.1 g of the sorbents was used. The volumes of the heavy metal ion solutions were identical to those described previously. The sorbents were shaken in the pH range of 2–6 in the 100 mg/L solutions. The parameters, including shaking speed of 180rpm, temperature of 293 K and time of 360 min, remained unchanged. After the effects of the optimal dose and pH value were estimated, the analyses of the effect of the phase contact time (1–360 min) and starting concentration of the solution (50–200 mg/L) were carried out at 180 rpm and a temperature of 293 K. The equilibrium sorption capacity (mg/g) and sorption percentage (%) for all the sorbents were estimated using the equations:(2)$$q_{e}=\frac{(C_{0}-C_{e})V}{m}$$
(3)$$\%S=\frac{\left(C_{0}-C_{e}\right)}{C_{0}}\cdot 100\%$$
where the starting and final concentrations of heavy metal ions [mg/L] are C_0_ and C_e_ [mg/L], respectively, the volume of the solution is denoted by the letter V [L], and m is the sorbent mass [g].

The kinetic parameters were computed using the pseudo-first order (PFO), pseudo-second order (PSO) and intra-particle diffusion (IPD) models based on the method above. The equilibrium analyses of the sorption process were carried out using 0.1 g of the sorbent and 20 mL of the solution in concentrations of 50–600 mg/L. The materials were shaken for 360 min in the same way as those in the sorption kinetics analysis. The Langmuir, Freundlich, Temkin and Dubinin–Raduszkiewicz isotherms were applied to derive the adsorption parameters based on the obtained data.

The experiments analyzing the effect of temperature were performed at temperatures of 293, 313 and 333 K with the same solution concentrations as those applied in the adsorption study. The thermodynamic parameters—that is, the changes in the standard free energy (ΔG°), standard enthalpy (ΔH°) and standard entropy (ΔS°)—were determined from the expressions:(4)$$\Delta G^{0^{}}=-RT\ln K_{d}$$
(5)$$\Delta G^{0}=\Delta H^{0}-T\Delta S^{0}$$
(6)$$K_{d}=\frac{C_{s}}{C_{e}}$$
(7)$$\ln K_{d}=\frac{\Delta H^{0}}{RT}+\frac{\Delta S^{0}}{R}$$
where R is the gas constant [J/mol·K], T is the temperature [K], K_d_ is the distribution coefficient, and the sorption capabilities of the adsorbent and adsorbate phases are C_s_ and C_e_ [mg/L], respectively.

The investigations were carried out in three replicates, and the errors were recorded. The graphs depict the results with the 7% margin of error.

### 2.3. Desorption Research

The impact of the kind of the desorbing agent should be assessed taking into consideration the reuse of the sorbents. Moreover, the analysis of the reversibility of the heavy metal ions on the tested materials in the adsorption process provides valuable information about the mechanism of the sorption process. Nitric, hydrochloric and sulfuric acids at the concentration of 1 mol/L were used for this research. Thus, the dried and weighed sorbents, after sorption (starting solution concentration 200 mg/L, time 360 min, pH 5, shaking speed 180 rpm and temperature 295 K), were shaken with the acidic agents for 360 min. The desorption results included the 7% measurement error and were obtained by threefold repetitions.

Using nitric acid as the most effective eluent, the washing efficiency of the heavy metal ions deposited on the sorbents was examined in terms of the phase contact time (1–360 min). The desorption percentage was calculated using the expression:(8)$$\%desorption=\frac{C_{des}}{C_{0}-C_{t}}100\%$$
where C_des_ is the quantity of heavy metal ions after the regeneration [mg/L].

## 3. Results and Discussion

### 3.1. Physicochemical Characterization of the Biochar-Based Materials

Table 1 presents the contents of ash, carbon, hydrogen and nitrogen in BC-CS 1-1, BC-CS 2-1 and BC-CS 4-1.

The high ash content of the biochar composites indicates the possibility of their use for agricultural purposes, because this accelerates plant growth. The highest percentage of nitrogen in BC-CS 1-1 relates to the fact that the highest amount of chitosan was added to this sorbent during the modification process. The SEM images of the biochar-based sorbents shown in Figure 2a–c present the heterogeneous, well-developed surfaces [25]. The chitosan particles deposited on the surface of the biochar are marked in the SEM images. These particles are unevenly distributed on the biochar surface due to the fact that biochar itself is a heterogeneous material. As the largest amount of chitosan was added to the BC-CS 1-1 sorbent during the synthesis, they are the most visible in that image.

Considering the amounts listed in Figure 3a–c compared to the specific surface area of the biochar, being 116 m^2^/g [26], only the BC-CS 4-1 sorbent had a larger specific surface area (134 m^2^/g). As for the chitosan-modified biochars BC-CS 1-1 and BC-CS 2-1, a great reduction in the specific surface area can be viewed as the result of the pore blockage depending on the amount of chitosan applied [5,27]. The large specific surface area of the BC-CS 4-1 may indicate a greater capacity for heavy metal ion removal as a result of the greater number of adsorption centers [28]. After the Cu(II) ion sorption, the BET specific surface area, as well as the total volume of pores and micropores of the chitosan-modified biochars, with the exception of BC-CS 1-1, increased. However, for all sorbents, the total size of the pores decreased compared to their values before the sorption process. It follows that, in the case of BC-CS 1-1, the Cu(II) ion sorption mechanism is the least effective. According to the IUPAC classification, the form of the adsorption/desorption isotherms of the sorbents fits the type-II class (Figure 3a–c). This kind of isotherm is found in sorbents with micro- and meso-pores [29]. Under lower relative pressures, the micropores are filled first, and the pressure increase results in the monolayer formation [30]. The central flattened part of the isotherm is called the BET region, and its high-pressure termination is due to the multilayer formation. The re-increase in pressure results in the capillary condensation of the mesopores [31]. At a relative pressure of 0.45, the adsorption and desorption curves are not reversible (not overlapping).

The pore size distribution of the chitosan-modified biochars obtained from the N_2_ adsorption isotherm curves is shown in Figure 4a–c, confirming the dominance of the micropores in the sorbent structure. The total pore volume values decreased with the increasing amount of chitosan added and amounted to 0.049, 0.07 and 0.094 cm^3^/g for BC-CS 1-1, 2-1 and 4-1, respectively. The BC-CS 1-1 sample had the highest amount of chitosan added. These results may indicate that the more chitosan one adds during the synthesis, the greater the blockage of the pores and, hence, the reduction in the pore volume will be.

As follows from the data in Table 2, silica is the crystalline phase with the largest content (ranging from 47.0 to 64.0%). The biochar-based sorbents also include calcium and magnesium carbonate, ranging from 3.2 to 15.0%; calcium carbonate, in a content range of 4.3–17.0%; sodium-aluminum silicate in the amount of 6.1–12.0%; potassium-aluminum silicate in the amount of 4.4–10.9%; and calcium chloride in a content range of 1.4–9.2%.

The XRD study shown in Figure 5a supports the existence of the crystalline phases specified in Table 2. Specifically, the peaks with the maximum intensities at approximately 2Θ = 26.5 and approx. 2Θ = 20.8 are related to silica [32]. The peaks at approx. 2Θ = 29.4 and 30.9 are related to the presence of carbon in the sorbents [33]. After the Cu(II) ions sorption, in all sorbents, we observed an additional phase due to the presence of copper hydroxychloride in a content range of 1.8–2.6%, indicating that Cu(II) ions were absorbed via precipitation, which resulted in the creation of sparingly soluble hydroxide. Figure 5b shows the percentage values of the substances in BC-CS 1-1.

The complex structures of the biochar-based sorbents can be seen in the ATR-FTIR spectra in Figure 6a–c.

The sorbents contain both aliphatic and aromatic groups [34]. The peaks in the 3900–3400 cm^−1^ range originate from the stretching vibrations of the hydroxyl groups [27]. In the range of 3400 to 3100 cm^−1^, peaks derived from the -NH groups of the amines are present [1]. The C-H stretching vibration peaks on the BC-CS 1-1 surface around 2850 cm^−1^ are connected to the acetylene polymerization [21,35]. The existence of C≡C and C≡N bonds or cumulative C=C=C and N=C=O double bonds is associated with the peaks in the range of 2000–2500 cm^−1^. At approx. 2000 cm^−1^, the peaks are related to the stretching vibrations of -CH_2_ groups in the long aliphatic chains [27]. The bands in the range of 1650–1500 cm^−1^ originate from the stretching vibrations of asymmetric carboxylate groups, and the peaks at around 1400 cm^−1^ can be attributed to the presence of stretching vibrations of symmetric carboxylate groups [36]. There are more bands in the region of 1012–1020 cm^−1^ that are attributable to the C-O-C groups vibrations [27,37]. The existence of bending vibrations beyond the plane of the C-H bond in the aromatic ring is demonstrated by the peaks in the region of 680–870 cm^−1^ [38]. The peaks associated with the hydroxyl and carboxyl groups are displaced after the Cu(II) ion sorption [39].

The thermogravimetric curves (Figure 7a–c) show that the chitosan-modified biochars were more thermally stable at lower temperatures, since the graph does not show a dramatic decline. With increasing temperature, the curves slope downward, indicating that the sorbents were less stable at higher temperatures. The first stage of weight loss in the temperature range 323–423 K is due to the loss of moisture and the volatile components contained in the sorbents.

The breakdown residues of hemicellulose, cellulose and lignin are responsible for the other phases of the biochar heat degradation, dropping to 823 K. The decomposition at higher temperatures results in the destruction of the carbon skeleton [40]. The total weight loss for BC-CS 1-1 was approx. 48%, for BC-CS 2-1 it was 40% and for BC-CS 4-1 it was 30%. Chitosan is less thermally stable than biochar [41], which explains the course of the TG curves of the chitosan-modified biochars, apart from BC-CS 4-1.

The temperatures at which the mass change is most noticeable can be determined from the DTG curves. The weight loss during heating is due to the water evaporation and elimination of the volatile components, as indicated by the peak emerging in the temperature range of 273–373 K. Moreover, the peaks observed in the range of 873–973 K result from the thermal decomposition of aromatic hydrocarbons and carbonates in the sorbents [40]. There is also a peak at approx. 553 K on the curves, which is related to the deacetylation and depolymerization of the chitosan on the biochar surfaces [42].

The pH at which the sorbent surface charge becomes zero is defined as the point of zero charge (pH_PZC_). The sorbent surface is negatively charged and can interact with positively charged ions at pH values greater than pH_PZC_. However, when the pH value is lower than the pH_PZC_ one, the sorbent surface is positively charged and can react with the negatively charged ions [29,43]. Figure 8a–c presents the pH_PZC_ values determined using the potentiometric method.

The pH_PZC_ values calculated using both methods differed slightly. The points of zero charge determined by the potentiometric method were 9.0, 8.3 and 8.6 for BC-CS 1-1, BC-CS 2-1 and BC-CS 4-1, respectively. The values obtained by the drift method were 8.8, 8.0 and 8.2 for BC-CS 1-1, BC-CS 2-1 and BC-CS 4-1, respectively. The heavy metal ion sorption took place in the pH range of 2–6. For each system, the pH value of the solution was lower than that of the pH_PZC_. In these systems, the sorbent surface was positively charged, a state which should lead to the electrostatic repulsion of heavy metal ions. Other interactions can sometimes be more powerful than the electrostatic forces, having minor impacts on the surface charge [44].

Based on the dependences of the surface charge density as a pH function (Figure 8a–c), it can be concluded that the surface charge density decreased with increasing solution pH. Furthermore, a negative surface charge was found at pH 9.2 for BC-CS 1-1, 9.5 for BC-CS 2-1 and 9.2 for BC-CS 4-1 at all the sodium chloride concentrations.

### 3.2. Dose Effect

The results of the effect of the sorbent mass and its dependence on the sorption process of the Cu(II) ions on the chitosan-modified biochars are shown in Figure 9a–c.

Following the experiment, which lasted 360 min, the values of the adsorbed ions amount were 13.82, 11.53 and 8.63 mg/g and, for the masses, 0.1 g, 0.15 g and 0.2 g, respectively, for BC-CS 1-1. For BC-CS 2-1, they were 14.48, 11.53 and 8.66 mg/g, and for the masses, 0.1 g, 0.15 g and 0.2 g, respectively. For BC-CS 4-1, they were 16.31, 11.50 and 8.79 mg/g, and for the masses, 0.1 g, 0.15 g and 0.2 g, respectively. The sorption percentages were 90.27%, 94.60% and 98.89% for BC-CS 1-1, BC-CS 2-1 and BC-CS 4-1, respectively (after 360 min reaction time). The results of the analysis of the effect of the sorbent mass depending on the phase contact time were obtained only for the Cu(II) ions. The reason for this is that the point tests of the influence of the mass on the sorption process of the remaining ions on all sorbents revealed that the ideal mass is 0.1 g; thus, this quantity of the sample was chosen for further investigations. The number of active centers accessible on the sample surfaces, theoretically, should increase with the increasing sorbent mass [45]. On the other hand, a decrease in the number of adsorbed ions with the increasing sorbent weight can be understood as the result of an insufficient amount of Cu(II) ions inhibiting the sorption process. Another reason may be the lack of saturation of the active centers due to the sorption process. Additionally, this effect can be explained as a result of the probable aggregation of the sorbent particles at a higher mass, reducing the dimensions of the specific surface area and the number of active sorbent centers on the sorbent surface [46].

### 3.3. The pH Effect

Figure 10a–c presents the results concerning the influence of the initial solution pH on the sorption of the Cu(II), Cd(II), Zn(II), Co(II) and Pb(II) ions using the biochar-based sorbents.

According to the findings, the number of adsorbed ions increased progressively with the increasing pH value in the case of all sorbents. For example, for the Cu(II) ions adsorbed on BC-CS 1-1, the values were 13.05, 17.71, 17.94 and 18.48 mg/g for pH 2, 3, 4 and 5, respectively. At pH 6, copper hydroxide precipitation was observed. For the other heavy metal ions, no precipitation of hydroxides was identified at pH 6, but a decrease in the q_t_ value compared to those obtained at pH 5 was observed. Therefore, the pH value of 5 was selected for further investigations of the influences of the phase contact time and the starting concentration on the sorption process effectiveness.

Taking the examined ions into account, the largest q_t_ values of 18.48, 18.57 and 18.61 mg/g for BC-CS 1-1, BC-CS 2-1 and BC-CS 4-1, respectively (at pH 5 after 360 min), were obtained for the Cu(II) ions. The Co(II) ions were removed from the aqueous solutions with the lowest efficiency. In the case of the metal ion solution at the pH value of 2, all the sorbents were characterized by the lowest adsorption capacity. The removal of smaller amounts of the above mentioned ions from the solution at low pH values may result from the competition between hydrogen (H^+^) and/or oxonium (H_3_O^+^) ions and heavy metal ions in the active sites of the adsorbents. Thus, when the H^+^ ion concentration increases, the surface of the sorbent becomes more positively charged, reducing the attraction between the sorbent and the metal [47]. When the pH value rises, the surface becomes negatively charged, thus facilitating a greater capture of metal ions from the aqueous media [48]. The metal solubility is reduced when the pH rises over 6, resulting in the precipitation of the metals as hydroxides.

### 3.4. Effects of the Phase Contact Time and Initial Concentration of the Solution

The results presented in Figure 11a–c and Appendix A confirm the statement that the effects of the Cu(II) sorption process and those of other heavy metal ions on the chitosan-modified biochars are related to the phase contact time and the starting concentration of the solution.

The number of adsorbed ions increased with the increasing contact time of the sorbent-solution phases and the concentration of the initial solution. The Cu(II) ion equilibrium capacities were found to be 7.87, 13.74, 15.40 and 21.95 mg/g for BC-CS 1-1; 7.95, 14.01, 18.22 and 24.53 mg/g for BC-CS 2-; and 9.18, 16.90, 17.79 and 25.51 mg/g for BC-CS 4-1, at the concentrations 50, 100, 150 and 200 mg/L, respectively. The largest number of adsorbed ions was found for BC-CS 4-1. The other values of q_e_ for the other heavy metal ions are listed in Table 3, Appendix A. The largest values of adsorbed Cd(II), Zn(II), Co(II) and Pb(II) ions were also obtained for BC-CS 4-1. As follows from the data in the above-mentioned tables, the affinity series of the sorbents for the tested ions is: Pb(II) > Cd(II) > Cu(II) > Zn(II) > Co(II) for BC-CS 1-1 and BC-CS 2-1, and Pb(II) > Cd(II) > Zn(II) > Cu(II) > Co(II) for BC-CS 4-1.

The data in Figure 11a–c show that the equilibrium was obtained faster at lower concentrations. Moreover, within 60 or 120 min, the equilibrium was established at the concentrations of 50 and 100 mg/L, while it was slower at the higher concentrations, i.e., 150 and 200 mg/L, lasting for 240 min. Furthermore, the shape of the curves indicates a two-stage sorption process. The process of the ions being absorbed in the solution took around 60 min in the first stage and between 60 and 360 min in the second stage. The equilibrium state was established in the second step. The two-stage sorption process can be explained by the accessibility of the free ion centers on the sorbent surface. At first, the number of accessible centers is very large, which results in the fast sorption process. Over time, the number of accessible sites decreases, which causes the curve to become flat until the equilibrium state is reached. Moreover, when approaching the equilibrium, repulsive forces act between the adsorbent and the adsorbate molecules, slowing down the ion sorption in the subsequent steps. Based on these findings, it can be stated that 360 min is a sufficient time for adsorption isotherm equilibrium investigations.

### 3.5. Adsorption Kinetics

Based on the predicted PFO model kinetic parameters (Table 3, Appendix A), it can be concluded that the smaller values of the determination coefficients and the poor adjustment of the q_1_ values to those obtained experimentally, q_exp_, do not enable the use of this model for the description of heavy metal ion sorption on the tested sorbents. The PSO model, with the determination coefficient R^2^ being greater than 0.988 for all the M(II) ions, provided the best fit with the experimental data. Moreover, the calculated values of q_2_ were very close to those obtained experimentally. This model follows the assumption that the rate limiting factor is the exchange or share of electrons between the adsorbent and the adsorbate [49]. Furthermore, the PSO model describes the processes defined by the ion exchange and chemisorption [37]. The values of the reaction rate coefficient, k_2_, range from 0.003 to 4.004. Additionally, it can be stated that the value of k_2_, in all systems, decreases with the increasing concentration of the starting solution, provided that the reaction rate decreases with the increasing concentration. It is worth adding, however, that the PSO model is not based on the reaction mechanism and demonstrates a fit over the entire time range in all adsorption systems, as confirmed by the previous studies [50].

In general, the adsorption rate increased with the increasing concentration of the solution used for the Cu(II), Cd(II), Zn(II) and Co(II) ions, which was confirmed by the increased the value of the parameter h with the increasing concentration. This is caused by the increase in the driving force required for the mass transfer, making it possible for more ions to reach the sorbent surface within a shorter period of time [51]. In the case of the Pb(II) ions, the h parameter decreased with the increasing solution concentration. This change can be attributed to the fact that Pb(II) ions are the most efficiently sorbed and, with these ions, the fastest equilibrium is achieved. Additionally, the sorption mechanism can undergo changes.

### 3.6. Temperature Effect, Adsorption and Thermodynamic Calculations

The determination of the influence of the adsorbed number of ions depending on the process temperature allows us to calculate thermodynamic parameters and determine whether a given process is exothermic or endothermic. As the data in Figure 12a–c and Appendix A show, the sorption process is temperature-dependent, and the q_e_ value rises with increasing temperature and reaches its maximum at 333 K. It is reasonable to predict that, as the temperature rises, the diffusion of ions to the sorbent surface will be enhanced, as will the number of active centers, resulting in the reaction efficiency enhancement. Similar to the sorption kinetic studies, BC-CS 4-1 proved to be the most effective sorbent. Considering the obtained results, the ions can be arranged into affinity series for each of the tested sorbents. For the chitosan-modified biochars at the mass ratio of 1-1, the series is as follows: Pb(II) > Cu(II) ≥ Cd(II) > Zn(II) > Co(II). At 2-1, the series is: Pb(II) > Cd(II) > Cu(II) > Zn(II) > Co(II). At 4-1, the series has the form: Pb(II) > Cu(II) > Zn(II) > Cd(II) > Co(II).

The results of the calculation of the equilibrium isotherms of the adsorption of the heavy metal ions are presented in Table 4, Appendix A, using the Langmuir, Freundlich, Temkin and Dubinin–Raduszkiewicz models. The Langmuir isotherm model (R^2^ > 0.904) offered the best fit with the experimental data, demonstrating the sorption process on the monolayer surface with no lateral interactions between the adsorbate ions [52]. Moreover, high values of the correlation coefficients were obtained in the case of the Freundlich isotherm. However, in the real systems, there was a limited fit between the experimental and theoretical data, as this isotherm is empirical. Since the Langmuir and Freundlich isotherms models did not sufficiently explain the adsorption mechanism, the Temkin and Dubinin–Raduszkiewicz isotherms were fitted to the experimental data. The adsorption parameters calculated using the Temkin model proved that the electrostatic interactions participate in the sorption process. The Dubinin–Raduszkiewicz model is applied for the determination of the adsorption mechanism on a heterogeneous surface based on the obtained activation energy value. For E_a_ in the range of 1–8 kJ/mol, the adsorption process is physical. The adsorption process occurs through the ion exchange when E_a_ is 8–16 kJ/mol. In the case when E_a_ ranges from 16 to 40 kJ/mol, the process is associated with chemisorption [53,54]. Taking into account the calculated activation energies (Table 4, Appendix A), it was stated that the removal of the M(II) ions from the aqueous media occurred through the ion exchange and chemisorption.

Thermodynamic parameters provide specific details regarding energy changes during adsorption. The heavy metal ion sorption on the chitosan-modified biochars was endothermic, as evidenced by the positive enthalpy values under the standard conditions (Table 5, Appendix A). The negative values of ΔS° confirm a decrease in the degree of system freedom, and the positive values indicate an enhancement of the randomness and disorder of the adsorbent surface after the sorption process [53,55]. The negative values of ΔG°, decreasing with the rising temperature, proved that the sorption process was spontaneous and favored a higher temperature [56].

The obtained values ranged from −11.04 to −18.07 kJ/mol, which corresponds to the range in which physisorption occurs (0 to −20 kJ/mol) [52]. The values of the decomposition coefficient (K_d_), which are usually close to one, increased with increasing temperature.

In Table 6, the adsorption capacities and/or sorption percentages obtained for different chitosan-modified biochars are listed. Because of the inclusion of various active functional groups, the inclusion of chitosan increased the surface adsorption sites of the biochars and the adsorption capacity of the composites. The results obtained for the BC-CS 1-1, 2-1 and 4-1 sorbents are similar to the results obtained by other authors. In the case of poly (acrylic acid)-grafted chitosan and biochar composites, higher q_t_ values were obtained, which meant that the modification process requires further optimization.

### 3.7. Sorption Mechanism

The physicochemical features of the investigated sorbents were evaluated using X-ray diffractometry and infrared spectroscopy before and after the Cu(II) sorption process in order to identify the possible mechanism of the M(II) ion sorption. Because of the nature of the adsorbent surface and possible interactions between the adsorbent and the adsorbate, the sorption mechanism of M(II) ions is complicated. The adsorption of copper ions on the chitosan-modified biochars takes place through the surface precipitation of sparingly soluble hydroxides, according to the results obtained using the XRD technique. This is confirmed by the presence of copper hydroxychloride, as one of the crystalline phases, as well as copper hydroxide. The surface precipitation is obtained from the equations:2Cu^2+^ + Cl^−^ + 3OH^−^ → Cu_2_Cl(OH)_3_
Cu^2+^ + 2OH^−^ → Cu(OH)_2_

Moreover, the process of the sorption of Cu(II) ions takes place by the ions’ reduction to Cu(I) and Cu(0). The ATR-FTIR spectra of the sorbents indicate the occurrence of surface functional groups, such as hydroxyl (-OH), carboxyl (-COOH) and phenolic (R-OH) groups. The previous research on the sorption of heavy metal ions on biochars revealed that the oxygen-containing groups play an important role in the sorption process [26]. The shift in the peak maxima after the sorption process provides evidence for the surface complexation with the participation of the Cu(II) ions [39]. The reaction of the Cu(II) ions with some surface groups of sorbents is demonstrated by the equations:-COOH + Cu^2+^ + H_2_O → -COOCu^+^ + H_3_O^+^
-OH + Cu^2+^ + H_2_O → -OCu^+^ + H_3_O^+^
R-OH + Cu^2+^ + H_2_O → R-OCu^+^ + H_3_O^+^

Aromatic structures are generated as a result of the thermal treatment of biomass in biochar due to the high degree of graphitization. Moreover, the π-π bond is one of the weak interactions observed in the compounds containing such structures. The π-π interaction mechanism is complex, and the aromatic structure of the high temperature biochar can act as an electron donor for coordination with the Cu(II) ions [63]. These interactions take place through the electron transfer from the donor to the acceptor.

The activation energies estimated using the Dubinin–Raduszkiewicz isotherm model proved that the removal of M(II) ions from the aqueous media takes place through the ion exchange and chemisorption. The K^+^, Na^+^ and Ca^2+^ ions in the biochars are exchanged with Cu^2+^ ions after the sorption process [32]. On the other hand, based on the calculated values of the Gibbs free energy under the standard conditions, it was assumed that the process is physically adsorptive—that is, it occurs due to the electrostatic interactions. This concept is further supported by the high values of the Temkin model determination coefficients and other isotherm parameters.

Generally, during the sorption (at pH 5) of heavy metals (M^2+^), there occurs a partial deprotonation of the functional groups, resulting in the replacement of hydrogen ions with heavy metal ions (Figure 13).

As follows from the analyses of the heavy metal ion sorption on chitosan, this process takes place due to the coordination of M(II) ions with the amine groups of chitosan:CS-NH_2_ + M^2+^ ⇄ [CS-NH_2_-M]^2+^
where M^2+^ is the cation of the metals [57].

In summary, it can be stated that the main mechanisms describing the heavy metal ion adsorption on the analyzed sorbents include complexation, electrostatic interactions, the ion exchange processes and surface precipitation.

### 3.8. Desorption Investigations

Considering the reuse of the sorbents as adsorption agents, the effect of the desorbing agent type should be investigated. The analysis of the reversibility of the M(II) ion adsorption process on the tested materials provides valuable information about the sorption process mechanism. After the sorption, the dried and weighed sorbents were stirred for 360 min with the acidic desorbing agents HNO_3_, HCl and H_2_SO_4_ at concentration of 1 mol/L (Figure 14a–c).

The preliminary results prove that HNO_3_ is the desorbing agent that most effectively leached the M(II) ions adsorbed on the above-mentioned sorbents. Comparing the desorption efficiency rates using hydrochloric and sulfuric acids, it can be stated that, in most cases, HCl was characterized by larger values of the desorption yield, with the exception of the Co(II) ions on BC-CS 2-1, where the sulfuric acid desorption efficiency was insignificant. The smallest efficiency of the elution of the Pb(II) ions was observed using H_2_SO_4_. This is due to the low solubility of lead sulphate, indicating that it should not be utilized for Pb(II) ion recovery. With the use of nitric acid for the Cu(II) ions, the highest desorption percentage was found for BC-CS 1-1 (92.67 ± 6.49%). The Cd(II) and Zn(II) ions were most easily washed from the BC-CS 4-1 surface, with the yields of 79.91 ± 5.60% and 88.97 ± 6.23%, respectively. In turn, the Co(II) and Pb(II) ions were the easiest to remove in the case of BC-CS 2-1, with the desorption percentages of 98.39 ± 6.89% and 99.13 ± 6.94%, respectively. As follows from the data, no regularity was observed in the sorption process, with the highest numbers of adsorbed ions observed for BC-CS 4-1.

Following the selection of nitric acid as the most effective desorbing agent, the washing efficiency of the heavy metal ions deposited on the sorbents was studied, taking the time of phase contact (1–360 min) into account. Figure 15a–c presents the results.

The desorption percentage increased with the increasing phase contact time until it reached a plateau. The obtained results allow us to determine the order of ion elution using HNO_3_: Pb(II) > Co(II) > Cu(II) > Zn(II) > Cd(II). According to these findings, acidic solutions can be utilized as excellent desorbing agents. In the case of solutions with a low pH value, the sorbent surface is protonated, thus enabling the positively charged ions to desorb. The incompatibility of the affinity of M(II) ions in the adsorption and desorption series proves that their removal on the tested sorbents was the result of more than one mechanism.

## 4. Conclusions

In order to make a comparison of the sorptive properties of the three types of the chitosan-modified biochars, BC-CS 1-1, BC-CS 2-1 and BC-CS 4-1, using the Cu(II), Cd(II), Zn(II), Co(II) and Pb(II) ions, their sorption performance was examined. The physicochemical analyses confirmed the well-developed micro- and meso-porous structures of the sorbent, in which silica is the main crystalline phase. Additionally, they exhibited the presence of surface functional groups, carboxyl, hydroxyl and phenol, determining the sorption mechanism, and confirmed the thermal stability of the materials. As follows from the results, operating parameters, such as the sorbent mass, solution pH, phase contact time, initial concentration of solution and temperature, determined the effectiveness of the removal process of the above-mentioned ions. Moreover, based on the PSO and Langmuir isotherm models, it can be observed that BC-CS 4-1 exhibited a greater affinity with heavy metal ions. This research allowed us to determine the series of the affinity of the ions: Pb(II) > Cd(II) > Cu(II) > Zn(II) > Co(II) for BC-CS 1-1 and BC-CS 2-1, and Pb(II) > Cd(II) > Zn(II) > Cu(II) > Co(II) for BC-CS 4-1. The Pb(II) and Cd(II) ions more readily formed surface complexes with the hydroxyl and carboxyl groups of biochar-based materials compared to the Co(II) ions. The parameters listed, using the Temkin model, confirmed the participation of electrostatic interactions in the sorption process, and the activation energies calculated using the Dubinin–Raduszkiewicz model prove that the sorption process takes place through ion exchange and chemisorption. The obtained thermodynamic parameters show that the process is endothermic, spontaneous and has the characteristic of physisorption. Based on the desorption tests, it was found that the acidic solutions can be used as effective desorbing agents, and the greatest percentages of desorption values were obtained using nitric acid. The best desorption efficiency was observed for Pb(II)-BC-CS 2-1 and the poorest for Cd(II)-BC-CS 1-1 (applying nitric acid). The differences in the affinity series for the adsorption and desorption confirm that different interactions are responsible for the process of removing heavy metal ions from aqueous media.

## Figures and Tables

**Figure 1 materials-15-06108-f001:**
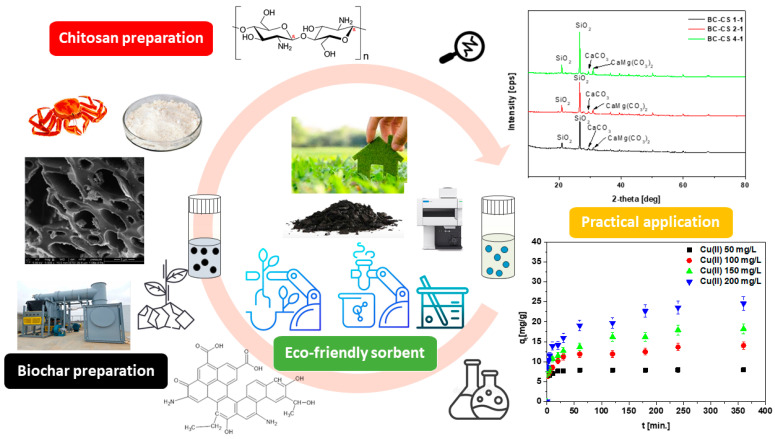
Synthesis scheme of the chitosan-modified biochar.

**Figure 2 materials-15-06108-f002:**
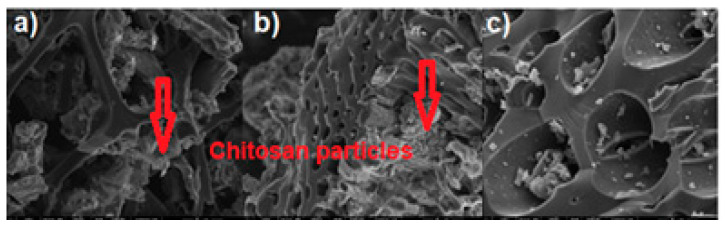
SEM images of (**a**) BC-CS 1-1, (**b**) BC-CS 2-1, (**c**) BC-CS 4-1 at 5000× magnification.

**Figure 3 materials-15-06108-f003:**
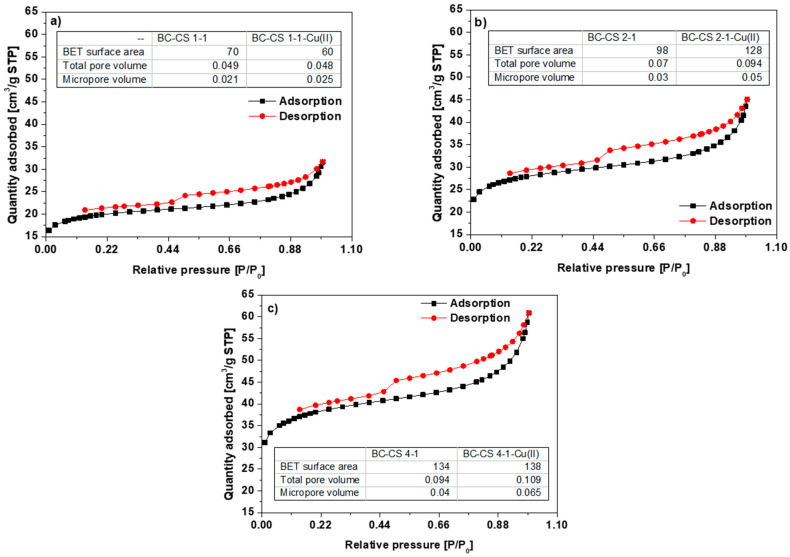
N_2_ adsorption/desorption isotherms of (**a**) BC-CS 1-1, (**b**) BC-CS 2-1 and (**c**) BC-CS 4-1.

**Figure 4 materials-15-06108-f004:**
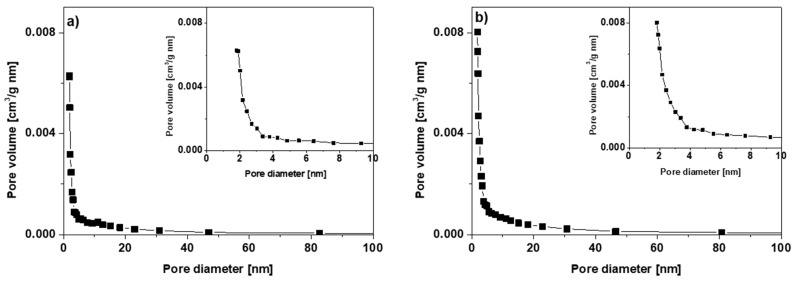
Pore size distribution of (**a**) BC-CS 1-1, (**b**) BC-CS 2-1 and (**c**) BC-CS 4-1.

**Figure 5 materials-15-06108-f005:**
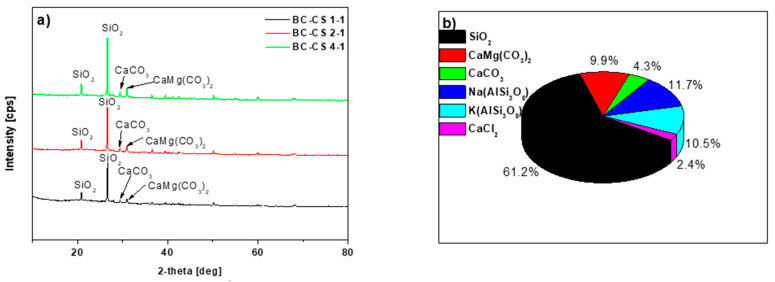
(**a**) XRD analysis of the chitosan-modified biochars, and (**b**) percentage values of the substances in BC-CS 1-1.

**Figure 6 materials-15-06108-f006:**
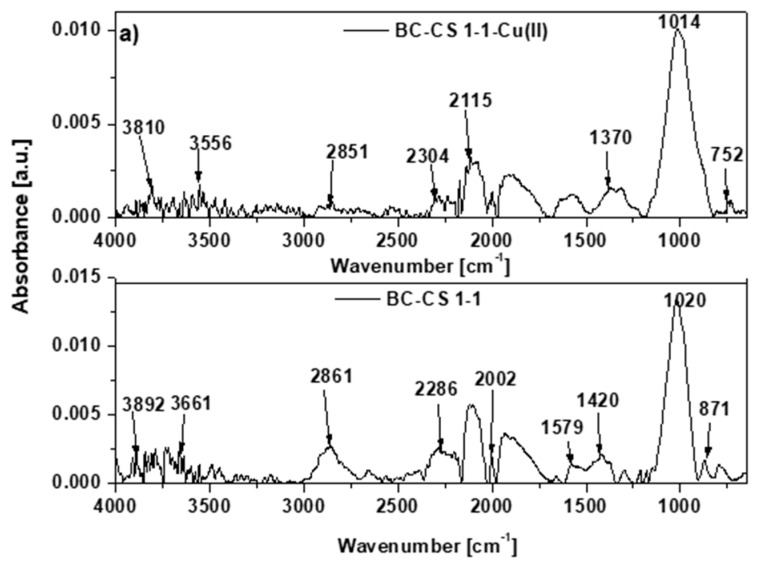
ATR–FTIR spectra of (**a**) BC-CS 1–1, (**b**) BC-CS 2–1 and (**c**) BC-CS 4–1 before and after the reaction with the Cu(II) ion solution.

**Figure 7 materials-15-06108-f007:**
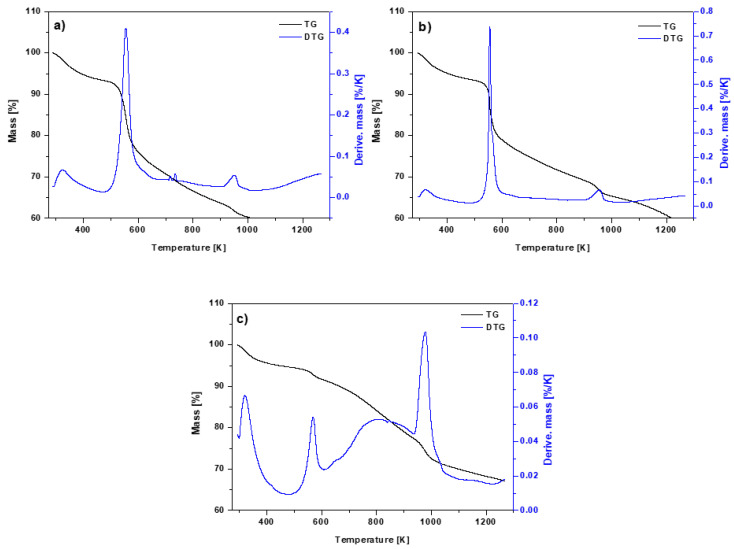
TG and DTG curves of (**a**) BC-CS 1-1, (**b**) BC-CS 2-1 and (**c**) BC-CS 4-1.

**Figure 8 materials-15-06108-f008:**
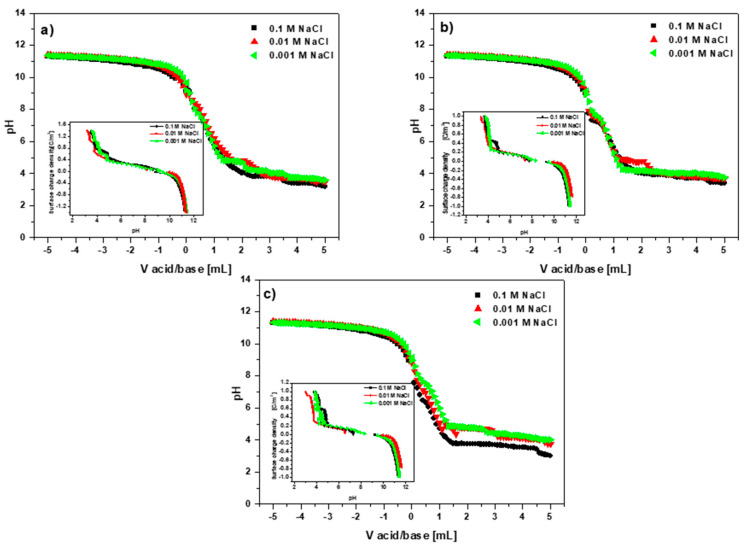
Point of zero charge and dependence of the surface charge density as a function of pH determined by the potentiometric method: (**a**) BC-CS 1–1, (**b**) BC-CS 2–1 and (**c**) BC-CS 4–1.

**Figure 9 materials-15-06108-f009:**
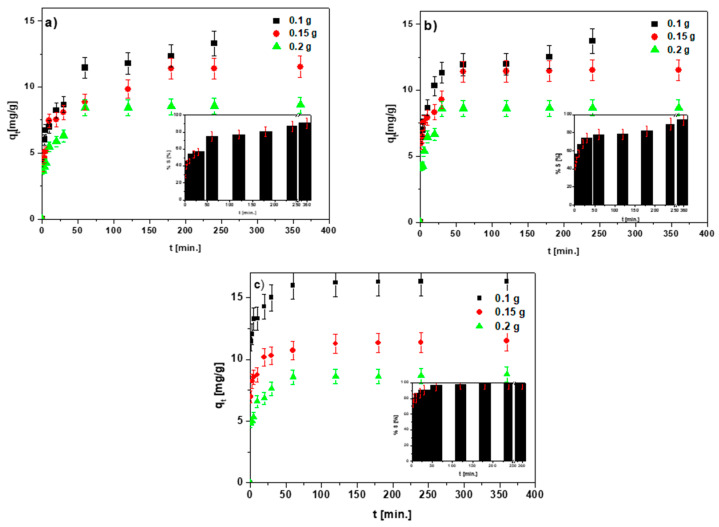
Effect of the sorbent mass on the Cu(II) ion sorption process depending on the phase contact time: (**a**) BC-CS 1-1, (**b**) BC-CS 2-1 and (**c**) BC-CS 4-1 (C_0_ 100 mg/L, pH 5, shaking speed 180 rpm, temperature 293 K).

**Figure 10 materials-15-06108-f010:**
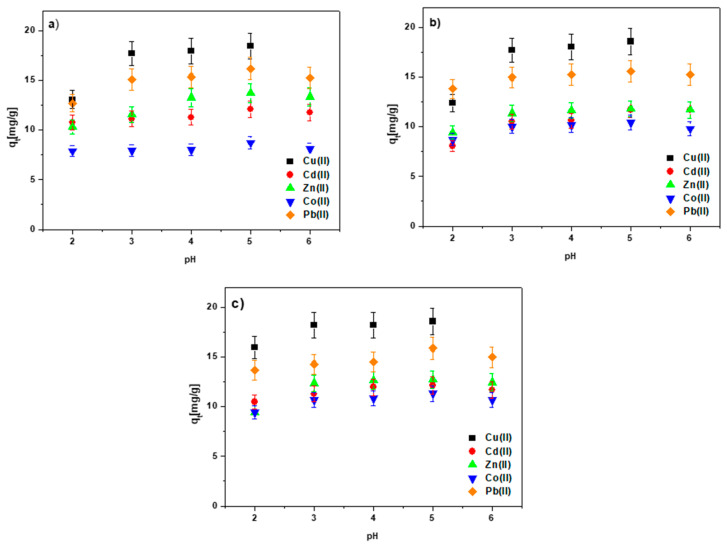
Effect of the pH solution on the heavy metal ion sorption on (**a**) BC-CS 1-1, (**b**) BC-CS 2-1 and (**c**) BC-CS 4-1 (C_0_ 100 mg/L, sorbent mass 0.1 g, pH 5, shaking speed 180 rpm, temperature 293 K).

**Figure 11 materials-15-06108-f011:**
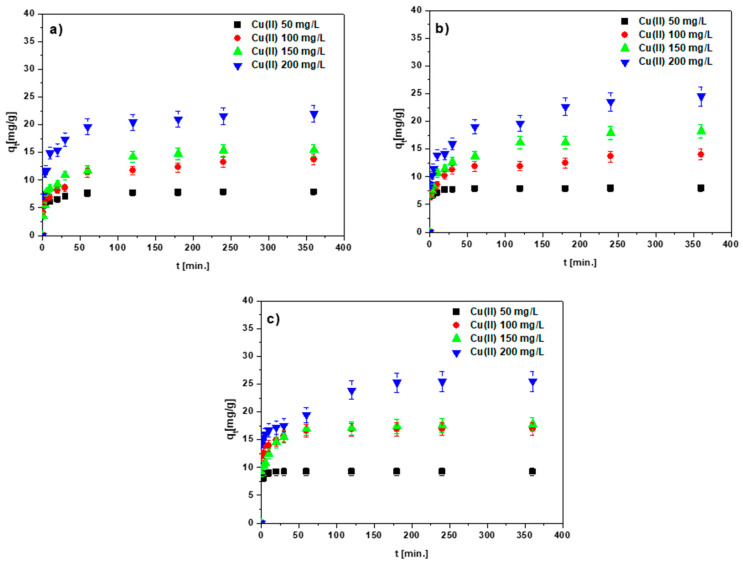
Effect of the phase contact time on the Cu(II) adsorption on (**a**) BC-CS 1-1, (**b**) BC-CS 2-1 and (**c**) BC-CS 4-1 (pH 5, sorbent mass 0.1 g, shaking speed 180 rpm, temperature 293 K).

**Figure 12 materials-15-06108-f012:**
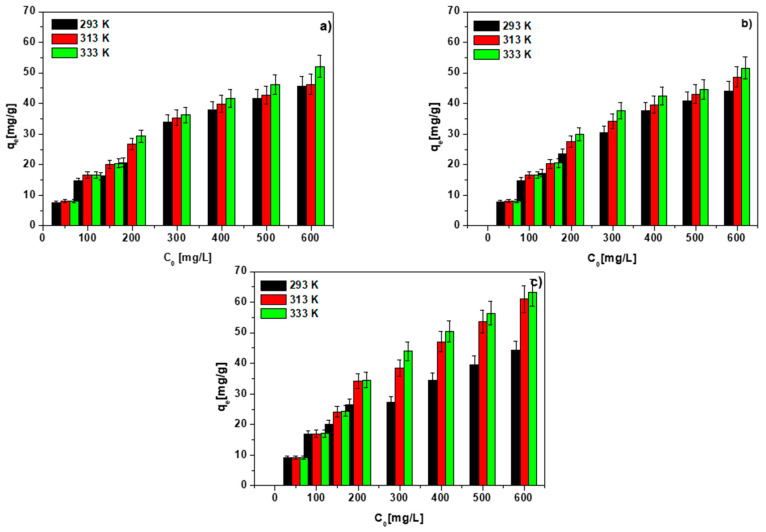
Effect of temperature on the Cu(II) sorption on (**a**) BC-CS 1-1, (**b**) BC-CS 2-1 and (**c**) BC-CS 4-1 (C_0_ 50–600 mg/L, sorbent mass 0.1 g, pH 5, shaking speed 180 rpm).

**Figure 13 materials-15-06108-f013:**
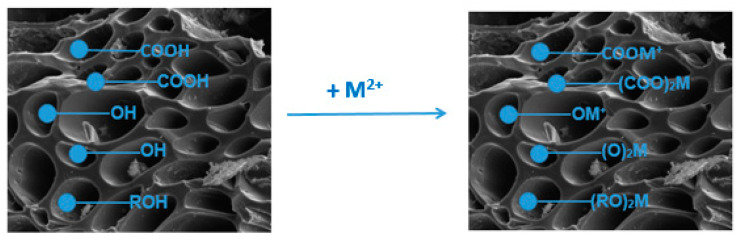
Scheme of the ion exchange on the biochar surface.

**Figure 14 materials-15-06108-f014:**
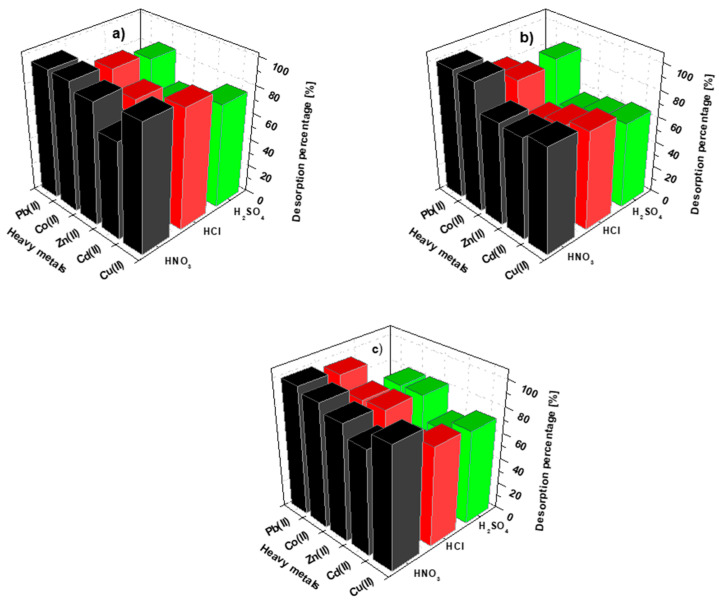
Elution of Cu(II), Cd(II), Zn(II), Co(II) and Pb(II) from the metal-loaded (**a**) BC-CS 1-1, (**b**) BC-CS 2-1 and (**c**) BC-CS 4-1, using HNO_3_, HCl and H_2_SO_4_ at a concentration 1 mol/L in three sorption–desorption cycles.

**Figure 15 materials-15-06108-f015:**
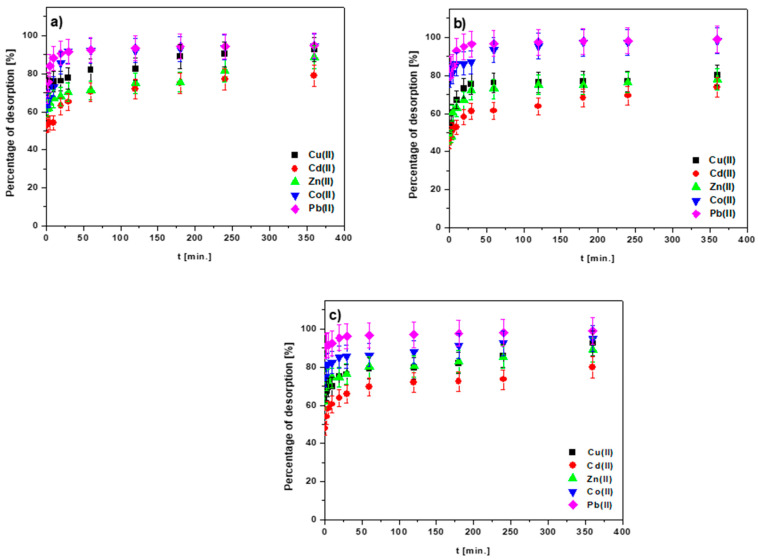
Effect of the phase contact time on the desorption percentages of the Cu(II), Cd(II), Zn(II), Co(II) and Pb(II) ions embedded on the surfaces of (**a**) BC-CS 1-1, (**b**) BC-CS 2-1 and (**c**) BC-CS 4-1, using HNO_3_ at a concentration of 1 mol/L.

**Table 1 materials-15-06108-t001:** Ash content and elemental analysis of BC-CS 1-1, BC-CS 2-1 and BC-CS 4-1.

Properties	BC-CS 1-1	BC-CS 2-1	BC-CS 4-1
Ash content	61.47	59.48	66.26
%C	40.17	41.42	39.56
%H	1.16	1.12	1.06
%N	1.53	1.14	0.96

**Table 2 materials-15-06108-t002:** Semi-quantitative analysis of BC-CS 1-1, BC-CS 2-1 and BC-CS 4-1 based on the XRD methods before and after the Cu(II) ion sorption.

Substances	BC-CS 1-1	BC-CS 2-1	BC-CS 4-1	BC-CS 1-1-Cu(II)	BC-CS 2-1- Cu(II)	BC-CS 4-1- Cu(II)
SiO_2_	61.2	60.5	59.0	64.0	52.0	47.0
CaMg(CO_3_)_2_	9.9	12.4	15.0	3.2	4.4	5.0
CaCO_3_	4.3	5.5	6.2	17.0	15.4	16.1
Na(AlSi_3_O_8_)	11.7	10.8	10.1	8.2	6.1	12.0
K(AlSi_3_O_8_)	10.5	8.7	8.0	4.4	10.9	9.0
CaCl_2_	2.4	2.1	1.6	1.4	9.2	8.3
Cu_2_Cl(OH)_3_	-	-	-	1.8	2.0	2.6

**Table 3 materials-15-06108-t003:** Parameters of the various adsorption kinetic models for the Cu(II), Cd(II), Zn(II) Co(II) and Pb(II) sorption on BC-CS 1-1.

Parameters											
C_0_ [mg/L]	q_exp_	PFO			PSO				IPD		
		$\log(q_{1}-q_{t})=\log(q_{1})-\frac{k_{1}t}{2.303}$			$\frac{t}{q_{t}}=\frac{1}{k_{2}q_{2}^{2}}+\frac{t}{q_{2}}$				qt=kit12+C		
		q_1_	k_1_	R^2^	q_2_	k_2_	h	R^2^	k_i_	C	R^2^
Cu(II)											
50	7.87	1.85	0.018	0.943	7.91	0.050	3.144	1.000	0.019	7.519	0.774
100	13.74	7.40	0.011	0.944	13.81	0.006	1.238	0.996	0.228	9.423	0.955
150	15.40	11.01	0.023	0.912	15.77	0.006	1.558	0.998	0.102	13.553	0.637
200	21.95	9.20	0.014	0.943	22.12	0.005	3.653	0.999	0.160	18.976	0.923
Cd(II)											
50	8.26	4.00	0.014	0.962	8.38	0.013	0.941	0.997	0.037	7.563	0.994
100	14.05	6.72	0.011	0.947	14.11	0.008	1.553	0.997	0.281	9.013	0.801
150	15.92	6.14	0.012	0.936	15.95	0.010	2.537	0.999	0.173	12.697	0.906
200	22.55	7.86	0.017	0.966	22.75	0.008	4.321	0.999	0.069	21.274	0.832
Zn(II)											
50	8.62	3.40	0.012	0.881	8.67	0.012	0.927	0.991	0.048	7.721	0.981
100	12.52	5.87	0.018	0.679	12.45	0.011	1.644	0.993	0.239	8.126	0.806
150	16.70	4.79	0.010	0.938	16.63	0.011	3.083	0.998	0.216	12.722	0.833
200	20.05	7.98	0.013	0.878	20.07	0.008	9.028	0.998	0.274	15.055	0.697
Co(II)											
50	5.17	1.48	0.007	0.811	5.09	0.038	0.993	0.998	0.069	3.832	0.897
100	7.92	2.34	0.009	0.914	7.86	0.024	1.462	0.998	0.095	6.104	0.968
150	11.11	5.15	0.009	0.901	11.03	0.009	1.038	0.992	0.204	6.260	0.805
200	12.02	4.18	0.006	0.959	11.75	0.011	1.476	0.995	0.280	6.698	1.000
Pb(II)											
50	8.58	0.03	0.017	0.660	8.58	4.004	294.494	1.000	0.001	8.569	0.964
100	16.64	0.08	0.025	0.598	16.65	1.336	370.191	1.000	0.001	16.641	0.857
150	21.49	2.05	0.037	0.953	21.53	0.080	37.316	1.000	0.001	21.484	0.957
200	30.12	5.75	0.022	0.943	30.27	0.019	17.399	1.000	0.018	29.782	0.995

**Table 4 materials-15-06108-t004:** Adsorption isotherm parameters and correlation coefficients for the adsorption of Cu(II), Cd(II), Zn(II), Co(II) and Pb(II) on BC-CS 1-1.

Isotherm Models	Parameters	Cu(II)	Cd(II)	Zn(II)	Co(II)	Pb(II)
Langmuir $q_{e}=\frac{q_{0}K_{L}C_{e}}{1+K_{L}C_{e}}$	q_e,exp_	45.76	48.56	44.10	24.42	68.36
	q_0_	52.35	96.63	48.91	27.38	65.13
	K_L_	0.024	0.004	0.020	0.007	0.050
	R^2^	0.944	0.978	0.966	0.963	0.948
Freundlich qe=KFCe1n	K_F_	5.37	0.97	5.34	1.45	9.37
	1/n	0.396	0.716	0.362	0.441	0.365
	R^2^	0.942	0.976	0.943	0.959	0.942
Temkin $q_{e}=\frac{RT}{b_{T}}\ln(K_{T}C_{e})$	K_T_	0.574	0.061	0.501	0.111	1.365
	b_T_	285.65	146.95	311.61	478.46	241.16
	R^2^	0.863	0.889	0.870	0.868	0.901
Dubinin–Raduszkiewicz $\ln q_{e}=\ln q_{m}-\beta \varepsilon^{2}$ $E_{a}=\frac{1}{\sqrt{2\beta }}$	q_m_	0.0015	0.0038	0.0012	0.0007	0.0020
	β	0.0041	0.0086	0.0038	0.0054	0.0036
	E_a_	16.086	8.642	11.423	9.603	11.739
	R^2^	0.928	0.968	0.915	0.944	0.957

**Table 5 materials-15-06108-t005:** Thermodynamic parameters for the sorption of Cu(II), Cd(II), Zn(II), Co(II) and Pb(II) ions on BC-CS 1-1.

Ions	K_d_			∆H^o^ [kJ/mol]	∆S^o^ [J/molK]	∆G^o^ [kJ/mol]		
	Temperature [K]					Temperature [K]		
	293	313	333			293	313	333
Cu(II)	0.185	0.190	0.243	5.38	4.05	−12.72	−13.65	−15.21
Cd(II)	0.138	0.144	0.148	1.55	−11.19	−11.99	−12.94	−13.84
Zn(II)	0.127	0.136	0.144	2.44	8.79	−11.81	−12.78	−13.75
Co(II)	0.062	0.069	0.083	5.92	−3.12	−10.04	−11.00	−12.22
Pb(II)	0.353	0.503	0.575	9.96	25.58	−14.29	−16.19	−17.59

**Table 6 materials-15-06108-t006:** Parallel results of the adsorption capacities/sorption percentages of metal ions sorbed on different types of chitosan-modified biochars.

Ions	Sorbent	Adsorption Capacity [mg/g]	Sorption Percentage [%]	References
Pb(II)	chitosan-modified bamboo biochar	-	52	[57]
Cd(II)		-	45	
Cu(II)		-	55	
Pb(II)	biochar supported zerovalent iron and chitosan (mass ratio 1:1:1)	-	100	[58]
Cr(VI)		-	35	
As(V)		-	60	
Pb(II)	chitosan- pyromellitic dianhydride modified biochar	8.62	-	[1]
Cd(II)		25.78	-	
Cu(II)		71.40	-	
Zn(II)	poly(acrylic acid) grafted chitosan and biochar composite	114.94	-	[59]
Cu(II)		111.11	-	
Co(II)		135.14	-	
Cd(II)	chitosan-modified kiwi branch biochar	126.58	-	[60]
Cd(II)	chitosan coated MgO-biochar	-	60	[32]
Pb(II)	chitosan-modified pine wood biochar	-	>70	[61]
Cr(VI)	magnetic BiFeO_3_ onto cross-linked chitosan	63.12	-	[62]

## Data Availability

Data are contained within the article and Appendix A.

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
