# Peer review of "Chitosan-Modified Biochars to Advance Research on Heavy Metal Ion Removal: Roles, Mechanism and Perspectives"

_materials, 2022, doi:10.3390/ma15176108_

Round 1
Reviewer 1 Report
1. The graphical figure need a high-resolution picture.
2. The researches of using biochar are many, they’d better to supply a dynamic investigation of the field.
3. Fig. 5 should be treated into two panels for better presentation.
4. Adding the refers of JCP, 2019, 1002; and JHM, 2019, 397, would be helpful.
5. From the author’s descriptions, the solution is very sensitive to pH. Could they control the solution pH using 1M acid and base accurately?
6. the initial adsorption rate H is used to describe the speed of adsorption. The value of H is also given in the table, but the text does not give the calculation formula of H, as well as the specific meaning and unit of each parameter. In addition, the author should discuss in detail the reasons why the initial adsorption rate of different heavy metal ions varies with solution concentration, especially for lead ions. And why the value of initial adsorption rate H decreases with the increase of solution concentration?
7. From the preparation scheme, the differences of the samples 1-1, 2-1, and 4-1 are the amount of biochar mixed. Why does the sample 3-1 with the amount of biochar 6g miss?
Author Response
- The graphical figure need a high-resolution picture.
Reply:
The graphical figure was corrected in the revised manuscript.
- The researches of using biochar are many, they’d better to supply a dynamic investigation of the field.
Reply:
The paper presents the investigations carried out using the static method based on the study of the influence of: sorbent mass, solution pH, phase contact time, initial solution concentration and temperature on the process effectiveness. Additionally, the desorption studies with the use of various acids are presented. Based on the obtained kinetic, adsorption and thermodynamic parameters, the sorption mechanism was proposed. The research is also supplemented with the results of the physicochemical analysis of the biochar based sorbents. The paper would be too extensive if a dynamic method was added.
- Fig. 5 should be treated into two panels for better presentation.
Reply:
Figure 5 was separator for better presentation.
- Adding the refers of JCP, 2019, 1002; and JHM, 2019, 397, would be helpful.
Reply:
The references were added.
- From the author’s descriptions, the solution is very sensitive to pH. Could they control the solution pH using 1M acid and base accurately?
Reply:
The pH of the solution was adjusted with 1M nitric acid or 1M sodium base. The solutions were added dropwise to reach the established pH values.
- the initial adsorption rate H is used to describe the speed of adsorption. The value of H is also given in the table, but the text does not give the calculation formula of H, as well as the specific meaning and unit of each parameter. In addition, the author should discuss in detail the reasons why the initial adsorption rate of different heavy metal ions varies with solution concentration, especially for lead ions. And why the value of initial adsorption rate H decreases with the increase of solution concentration?
Reply:
The equation for the initial adsorption rate [mg/g∙min] is as follows:
h=k2∙q22
where: k2 - the reaction rate constant of the PSO model [g/mg∙min], q2 - the amount of metal ions adsorbed at equilibrium [mg/g].
In general, the adsorption rate increases with the increasing concentration of the solution for Cu(II), Cd(II), Zn(II) and Co(II) ions, which is confirmed by increase the value of the parameter h with the increasing concentration. This is caused by the increase in the driving force for the mass transfer, making it possible for more ions to reach the sorbents surface in a shorter period of time [1]. In the case of Pb(II) ions, the h parameter decreases with the increasing solution concentration. This change can be due to the fact that Pb(II) ions are the best sorbed and the fastest equilibrium is achieved. Additionally, the sorption mechanism can undergo change.
- From the preparation scheme, the differences of the samples 1-1, 2-1, and 4-1 are the amount of biochar mixed. Why does the sample 3-1 with the amount of biochar 6g miss?
Reply:
The synthesis of the chitosan modified biochars at the ratios 1-1, 2-1 and 4-1 was assumed earlier. Another paper (in the review) also presents the studies for BC-CS 8-1. The research plan and the obtained sorbents were compared in terms of the effectiveness of heavy metal ions removal.
Reviewer 2 Report
Dear authors
Manuscript deals about an interesting topic. However, the quality is not enough for their publication. First of all, the English needs to be revised. Additionally, there are some mistakes or sentences that needs to be revised.
For example,
Introduction:
Line 28. Not all metals are carcinogenic and it depends on concentration.
Line 43. Not all biochars have large specific surface area
Line 46. What is active carbon? Activated carbon?
Lines 104. Why there are this variation between 4 an 25mg? The TG will depend on the quantity used. So, it is recommended to use similar weights
Line 129. Authors said that to achieve the desired pH they add 1mL of HNO3 or NaOH. However, in this case it is possible to obtained a desired pH??? What concentration? In general, authors need to improve the description of methodology
Figure 2. Please, indicate the particles
Authors perform discussion of data that are not present in the Tables or Figures. For example, the surface area.
In Lines 205-207, authors said " After the Cu(II) ions sorption, the BET specific surface area as well as the total volume of pores and that of micropores of chitosan modified biochars, apart from BC-CS 1-1, increased". Where are these data?
Additionally authors need to show data of pristine biochar in order to compare with that obtained after chitosan addition.
Characterization of biochar is to scarce. Authors need to perform a wide characterization including elemental analysis and ash content. Also, the pristine biochar needs to be characterized in order to compare with that biochars modified with chitosan.
Data needs to be re-organized. Figura 6 include FTIR of biochar after Cu adsorption. First it is necessary to show and discuss the characterization of biochars used.
Lines 342-345. Authors said "The thermogravimetric curves (Figure 7a-c) show that the chitosan modified biochars are more thermally stable at lower temperatures since the graph does not show a dramatic decline" This sentence is comparing to pristine biochar? Where are thermogravimetric curves of biochar without chitosan addition?
Author Response
Line 28. Not all metals are carcinogenic and it depends on concentration.
Reply:
The statement was changed.
Modification:
These substances are dangerous for the health and life of humans and animals due to the tendency towards bioaccumulation and some of them being carcinogenic.
Line 43. Not all biochars have large specific surface area
Reply:
The statement was changed.
Modification:
The stability, porosity and presence of the functional groups of surface are obvious advantages of the biochars.
Line 46. What is active carbon? Activated carbon?
Reply:
As a matter of fact, activated carbon should be used.
Modification:
As compared to the activated carbons, biochars are cost-effective sorbents.
Lines 104. Why there are this variation between 4 an 25mg? The TG will depend on the quantity used. So, it is recommended to use similar weights
Reply:
The sample weight used for the thermogravimetric test was examined.
Modification:
The test sample, weighing about 25 mg, was heated at a constant rate of 10 K/min in the temperature range of 298-1273 K in the nitrogen atmosphere.
Line 129. Authors said that to achieve the desired pH they add 1mL of HNO3 or NaOH. However, in this case it is possible to obtained a desired pH??? What concentration? In general, authors need to improve the description of methodology
Reply
The statement was changed.
Modification:
To achieve the desired pH, 1 mol/L HNO3 and/or 1 mol/L NaOH were added dropwise to the solution.
Figure 2. Please, indicate the particles
Reply:
The chitosan particles deposited on the surface of biochar are marked in the SEM images. These particles are unevenly distributed on the biochar surface, due to the fact that biochar itself is a heterogeneous material. Due to the largest weight of the added chitosan to the BC-CS 1-1 sorbent during the synthesis, they are the most visible. All changes were marked in Fig.2.
Authors perform discussion of data that are not present in the Tables or Figures. For example, the surface area.
Reply:
The specific surface area data are given in the table in Figure 5a-c.
In Lines 205-207, authors said " After the Cu(II) ions sorption, the BET specific surface area as well as the total volume of pores and that of micropores of chitosan modified biochars, apart from BC-CS 1-1, increased". Where are these data?
Reply:
The BET surface area, pore diameter, total pore volume and micropore volume data are presented in the tables in Figure 5a-c.
Additionally authors need to show data of pristine biochar in order to compare with that obtained after chitosan addition.
Reply:
The data for pristine biochar are presented in [26]. Comparing the qt values of pristine biochar and chitosan modified biochars, it can be concluded that pristine biochar showed greater efficiency in heavy metal ions removal. It can be related to the biochar pores blockage by the chitosan addition. Hence, the synthesis process requires further optimization.
Characterization of biochar is to scarce. Authors need to perform a wide characterization including elemental analysis and ash content. Also, the pristine biochar needs to be characterized in order to compare with that biochars modified with chitosan.
Reply
The elemental analysis of pristine biochar was presented in [26]. For chitosan modified biochars elemental analysis has not been performed. This analysis will be included in the next paper.
Data needs to be re-organized. Figura 6 include FTIR of biochar after Cu adsorption. First it is necessary to show and discuss the characterization of biochars used.
Reply:
Figure 6 consists of two panels - at the bottom - biochar before the sorption process, and at the top - biochar after the Cu(II) ions sorption process. The characteristics of biochar before and after sorption have been discussed in the paper.
Lines 342-345. Authors said "The thermogravimetric curves (Figure 7a-c) show that the chitosan modified biochars are more thermally stable at lower temperatures since the graph does not show a dramatic decline" This sentence is comparing to pristine biochar? Where are thermogravimetric curves of biochar without chitosan addition?
Reply:
The thermogravimetric curves of pristine biochar were presented in [26]. In that paper, on the basis of the TG research, it was found that the total loss of biochar mass up to the temperature of 1273 K is about 35%. Comparing the percentage of weight loss for chitosan modified biochars: 48% for BC-CS 1-1, 40% for BC-CS 2-1 and 30% for BC-CS 4-1, it was found that the greater thermal stability was reached by BC-CS 4-1.
Reviewer 3 Report
The work by D. Kołodyńska is indeed a very interesting and well-written manuscript. I have a few questions about the data.
1. I did not find the recyclability study? is this material not recyclable?
2. When desorption is performed, this might also leach out some of the adsorbed chitosan. Can you determine how many mmol of chitosan is desorbed?
3. Figures quality needs to improve.
4. Major claim of such work is always about extracting heavy metal ions from the waste stream; however, no one actually does such a study. Can the author perform such an experiment and show real use of this development?
5. Can authors compare their work with different chitosan composite materials prepared and published before?
6. Can the author highlight which group is more important (OH or NH2 or both)? Can the author perform additional studies to protect OH and NH2 separately and compare all three results? If NH2 is only important why OH is required?
Although, the work is comparative, competitive and publishable to the this journal.
Author Response
1.I did not find the recyclability study? is this material not recyclable?
Reply:
The recyclability study was presented in Subsection 3.7. The desorption investigations and more specifically in Fig.14a-c with the caption: Elution of Cu(II), Cd(II), Zn(II), Co(II) and Pb(II) from metal-loaded (a) BC-CS 1-1, (b) BC-CS 2-1 and (c) BC-CS 4-1 using HNO3, HCl and H2SO4 at the concentration 1 mol/L in three sorption-desorption cycles. However, no significant change in the sorption/ desorption efficiency was observed. Therefore, they can be reused in the acid treatment without their adsorption capacity loss.
- When desorption is performed, this might also leach out some of the adsorbed chitosan. Can you determine how many mmol of chitosan is desorbed?
Reply:
The chitosan leaching were not carried out after the desorption proces.
- Figures quality needs to improve.
Reply:
All figures were made using the Origin8.6 software.
- Major claim of such work is always about extracting heavy metal ions from the waste stream; however, no one actually does such a study. Can the author perform such an experiment and show real use of this development?
Reply:
The paper presents the model studies of heavy metal ions removal taking into account the influence of the sorbent mass, solution pH, phase contact time and initial solution concentration as well as temperature. These studies are an introduction to more extensive studies on the real systems - sewages. The results of this research will be presented in the next paper.
- Can authors compare their work with different chitosan composite materials prepared and published before?
Reply:
Table 5 presents the adsorption capacities and/or sorption percentages obtained for different chitosan composite materials. The results obtained for the BC-CS 1-1, 2-1 and 4-1 sorbents are similar to those obtained by the other authors. In the case of poly (acrylic acid) grafted chitosan and biochar composite, there were obtained larger qt values, which indicates that the modification process requires further optimization.
- Can the author highlight which group is more important (OH or NH2 or both)? Can the author perform additional studies to protect OH and NH2 separately and compare all three results? If NH2 is only important why OH is required?
Reply:
Which group: -OH or –NH2 is more important was not studied. However, the peaks of the stretching -OH groups vibrations appearing in the range of 3900-3400 cm-1 can indicate the presence of hydrogen bonds as one of the interactions in the sorbents [2]. Nevertheless, as it results from the studies of heavy metal ion sorption on chitosan, this process occurs due to the coordination of M(II) ions with the amine groups of chitosan.
- Demirbas, E.; Nas, M.Z. Batch Kinetic and Equilibrium Studies of Adsorption of Reactive Blue 21 by Fly Ash and Sepiolite. DES 2009, 243, 8–21, doi:10.1016/j.desal.2008.04.011.
- Gong, X.; Lu, H.; Li, K.; Li, W. Effective Adsorption of Crystal Violet Dye on Sugarcane Bagasse – Bentonite / Sodium Alginate Composite Aerogel : Characterisation , Experiments , and Advanced Modelling. Sep. Purif. Technol. 2022, 286, 120478.
Round 2
Reviewer 2 Report
Dear authors,
The characterization of modified biochar is to low. Please, include elemental analysis and ash content. The reason can´t be that you are going to use data in a next article.
I continues to think that authors need to perform a wide review of english
Author Response
Please see the attachement.
